# Self-Adaptive Training: beyond Empirical Risk Minimization

**Lang Huang**
Peking University
laynehuang@pku.edu.cn

**Chao Zhang***
Peking University
c.zhang@pku.edu.cn

**Hongyang Zhang***
TTIC
hongyanz@ttic.edu

## Abstract

We propose self-adaptive training—a new training algorithm that dynamically calibrates training process by model predictions without incurring extra computational cost—to improve generalization of deep learning for potentially corrupted training data. This problem is important to robustly learning from data that are corrupted by, e.g., random noise and adversarial examples. The standard empirical risk minimization (ERM) for such data, however, may easily overfit noise and thus suffers from sub-optimal performance. In this paper, we observe that model predictions can substantially benefit the training process: self-adaptive training significantly mitigates the overfitting issue and improves generalization over ERM under both random and adversarial noise. Besides, in sharp contrast to the recently-discovered double-descent phenomenon in ERM, self-adaptive training exhibits a single-descent error-capacity curve, indicating that such a phenomenon might be a result of overfitting of noise. Experiments on the CIFAR and ImageNet datasets verify the effectiveness of our approach in two applications: classification with label noise and selective classification. The code is available at https://github.com/LayneH/self-adaptive-training.

## 1 Introduction

Empirical Risk Minimization (ERM) has received significant attention due to its impressive generalization in various fields [36, 16]. However, recent works [46, 23] cast doubt on the traditional views on ERM: techniques such as uniform convergence might be unable to explain the generalization of deep neural networks, because ERM easily overfits training data even though the training data are partially or completely corrupted by random noise.

To take a closer look at this phenomenon, we evaluate the generalization of ERM on the CIFAR10 dataset [18] with 40% of data being corrupted at random (see Section 2 for details). Figure 1a displays the accuracy curves of ERM that are trained on the noisy training sets under four kinds of random corruptions: ERM easily overfits noisy training data and achieves nearly perfect training accuracy. However, the four subfigures exhibit very different generalization behaviors which are indistinguishable by the accuracy curve on the noisy training set on its own.

Despite a large literature devoted to analyzing the phenomenon either in the theoretical or empirical manners, many fundamental questions remain unresolved. To name a few, the work of [46] showed that early stopping can improve generalization. On the theoretical front, the work of [19] considered the label corruption setting, and proved that the first few training iterations fits the correct labels and overfitting only occurs in the last few iterations: in Figure 1a, the accuracy increases in the early stage and the generalization errors grow quickly after certain epochs. Admittedly, stopping at early epoch improves generalization in the presence of label noise (see the first column in Figure 1a); however, it

---

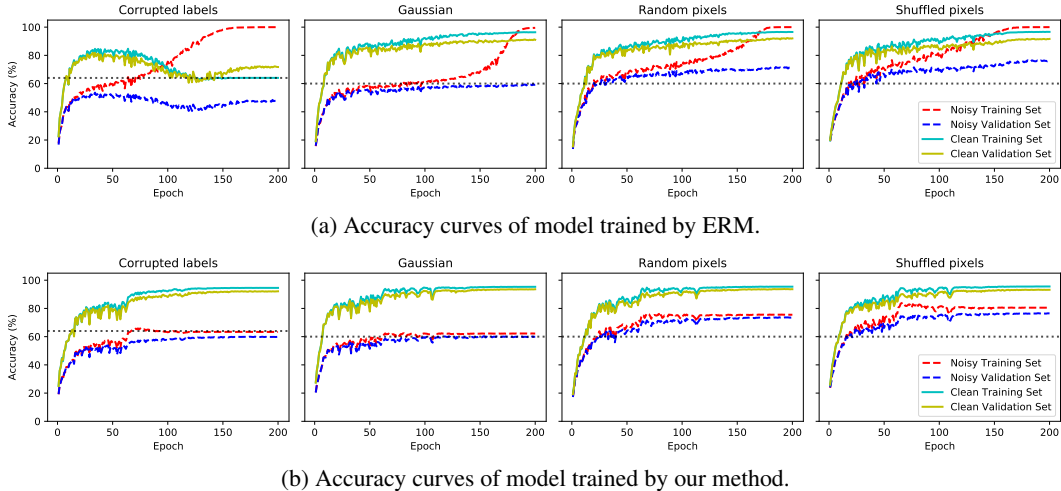

(a) Accuracy curves of model trained by ERM.

(b) Accuracy curves of model trained by our method.

Figure 1: Accuracy curves of model trained on noisy CIFAR10 training set (corresponding to the red dashed curve) with 40% corrupted data. The horizontal dotted line displays the percentage of clean data in the training sets.

remains unclear how to properly identify such an epoch. Moreover, the early-stop mechanism may significantly hurt the performance on the clean validation sets, as we can see in the second to the fourth columns of Figure 1a.

Our work is motivated by this fact and goes beyond ERM. We begin by making the following observations in the leftmost subfigure of Figure 1a: the peak of accuracy curve on the clean training set (80%) is much higher than the percentage of clean data in the noisy training set (60%). This finding was also previously reported by [33, 15, 19] under label corruption and suggested that model predictions might be able to magnify useful underlying information in data. We confirm this finding and show that the pattern occurs under various kinds of corruptions more broadly (see Figure 1a). We thus propose *self-adaptive training*, a carefully designed approach which dynamically uses model predictions as a guiding principle in the design of training algorithm. Figure 1b shows that our approach significantly alleviates the overfitting issue on the noisy training set, reduces the generalization error on the corrupted distributions, and improves the performance on the clean data.

## 1.1 Summary of our contributions

Our work sheds light on understanding generalization of deep neural networks under noise.

- We analyze the standard ERM training process of deep networks on four kinds of corruptions (see Figure 1a). We describe the failure scenarios of ERM and observe that useful information for classification has been distilled to model predictions in the first few epochs. This observation motivates us to propose self-adaptive training for robustly learning under noise.
- We show that self-adaptive training improves generalization under both label-wise and instance-wise random noise (see Figures 1 and 2). Besides, self-adaptive training exhibits a single-descent error-capacity curve (see Figure 3). This is in sharp contrast to the recently-discovered double-descent phenomenon in ERM which might be a result of overfitting of noise.
- While adversarial training may easily overfit adversarial noise, our approach mitigates the overfitting issue and improves adversarial accuracy by ∼3% over the state-of-the-art (see Figure 4).

Our approach has two applications and advances the state-of-the-art by a significant gap.

- Classification with label noise, where the goal is to improve the performance of deep networks on clean test data in the presence of training label noise. On the CIFAR datasets, our approach achieves up to 9.3% absolute improvement on the classification accuracy over the state-of-the-art. On the ImageNet dataset, our approach improves over ERM by 2% under 40% noise rate.
- Selective classification, which aims to trade prediction coverage off against classification accuracy. Our approach achieves up to 50% relative improvement over the state-of-the-art on two datasets.

**Differences between our methodology and existing works on robust learning** Self-adaptive training consists of two components: a) the moving-average scheme that *progressively* corrects problematic labels using model predictions; b) the re-weighting scheme that *dynamically* puts less weights on the erroneous data. With the two components, our algorithm is robust to both instance-wise and label-wise noise, and is ready to combine with various training schemes such as natural and adversarial training, without incurring multiple rounds of training. In contrast, a vast majority of works on learning from corrupted data follow a preprocessing-training fashion with an emphasis on the label-wise noise only: this line of research either discards samples based on disagreement between noisy labels and model predictions [7, 6, 50, 25], or corrects noisy labels [4, 40]; [41] investigated a more generic approach that corrects both label-wise and instance-wise noise. However, their approach inherently suffers from extra computational overhead. Besides, unlike the general scheme in robust statistics [34] and other re-weighting methods [17, 32] that use an additional optimization step to update the sample weights, our approach directly obtains the weights based on accumulated model predictions and thus is much more efficient.

## 2 Improved Generalization of Deep Networks

### 2.1 Preliminary

In this section, we conduct the experiments on the CIFAR10 dataset [18], of which we split the original training data into a training set (consists of first 45,000 data pairs) and a validation set (consists of last 5,000 data pairs). We consider four random noise schemes according to [46], where the data are *partially* corrupted with probability $p$: 1) *Corrupted labels*. Labels are assigned uniformly at random; 2) *Gaussian*. Images are replaced by random Gaussian samples with the same mean and standard deviation as the original image distribution; 3) *Random pixels*. Pixels of each image are shuffled using independent random permutations; 4) *Shuffled pixels*. Pixels of each image are shuffled using a fixed permutation pattern. We consider the performance on both the noisy and the clean sets (i.e., the original uncorrupted data), while the models can only have access to the noisy training sets.

**Notations** We consider $c$-class classification problem and denote the images by $\boldsymbol{x}_i \in \mathbb{R}^d$, labels by $\boldsymbol{y}_i \in \{0, 1\}^c, \boldsymbol{y}_i^\mathsf{T} \mathbf{1} = 1$. The images $\boldsymbol{x}_i$ or labels $\boldsymbol{y}_i$ might be corrupted by one of the four schemes we have described. We denote the logits of the classifier (e.g., parameterized by a deep network) by $f(\cdot)$.

### 2.2 Our approach: Self-Adaptive Training

To alleviate the overfitting issue of ERM in Figure 1a, we present our approach to improve the generalization of deep networks on the corrupted data.

**The blessing of model predictions** As a warm-up algorithm, a straight-forward way to incorporate model predictions into the training process is to use a convex combination of labels and predictions as the training targets. Concretely, given data pair $(\boldsymbol{x}_i, \boldsymbol{y}_i)$ and prediction $\boldsymbol{p}_i = \mathrm{softmax}(f(\boldsymbol{x}_i))$, we consider the training target $\boldsymbol{t}_i = \alpha \times \boldsymbol{y}_i + (1 - \alpha) \times \boldsymbol{p}_i$, where $\boldsymbol{t}_i \in [0, 1]^c, \boldsymbol{t}_i^\mathsf{T} \mathbf{1} = 1$. We then minimize the cross entropy loss between $\boldsymbol{p}_i$ and $\boldsymbol{t}_i$ to update the classifier $f$ in each training iteration. However, this naive algorithm suffers from multiple drawbacks: 1) model predictions are inaccurate in the early stage of training, and may be unstable in the presence of regularization such as data augmentation. This leads to instability of $\boldsymbol{t}_i$; 2) this scheme can assign at most $1 - \alpha$ weight on the true class when $\boldsymbol{y}_i$ is wrong. However, we aim to correct the erroneous labeling. In other words, we expect to assign nearly 100% weight on the true class.

To overcome the drawbacks, we use the accumulated predictions to augment the training dynamics. Formally, we initialize $\boldsymbol{t}_i \leftarrow \boldsymbol{y}_i$, fix $\boldsymbol{t}_i$ in the first $\mathrm{E}_s$ training epochs, and update $\boldsymbol{t}_i \leftarrow \alpha \times \boldsymbol{t}_i + (1 - \alpha) \times \boldsymbol{p}_i$ in each following training epoch. The exponential-moving-average scheme alleviates the instability issue of model predictions, smooths out $\boldsymbol{t}_i$ during the training process and enables our algorithm to completely change the training labels if necessary. Momentum term $\alpha$ controls the weight on the model predictions. The number of initial epochs $\mathrm{E}_s$ allows the model to capture informative signals in the data set and excludes ambiguous information that is provided by model predictions in the early stage of training.

**Sample re-weighting** Based on the scheme presented above, we introduce a simple yet effective sample re-weighting scheme on each sample. Concretely, given training target $\boldsymbol{t}_i$, we set $w_i = \max_j \boldsymbol{t}_{i,j}$. The sample weight $w_i \in [\frac{1}{c}, 1]$ reveals the labeling confidence of this sample. Intuitively,

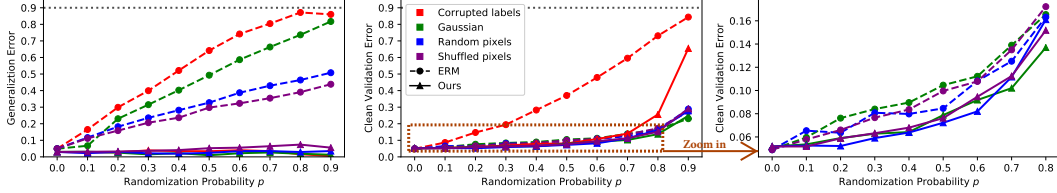

Figure 2: Generalization error and clean validation error under four kinds of random noise (represented by different colors) for ERM (the dashed curves) and our approach (the solid curves) on CIFAR10. We zoom-in the dashed rectangle region and display it in the third column for clear demonstration.

all samples are treated equally in the first $E_s$ epochs. As target $t_i$ being updated, our algorithm pays less attention to potentially erroneous data and learns more from potentially clean data. This scheme also allows the corrupted samples to re-attain attention if they are confidently corrected.

**Putting everything together**    We use stochastic gradient descent to minimize:

$$\mathcal{L}(f) = -\frac{1}{\sum_i w_i} \sum_i w_i \sum_j \boldsymbol{t}_{i,j} \log \boldsymbol{p}_{i,j} \tag{1}$$

during the training process. Here, the denominator normalizes per sample weights and stabilizes the loss scale. We name our approach *Self-Adaptive Training* and display the pseudocode in Algorithm 1. We fix the hyper-parameters $E_s = 60$, $\alpha = 0.9$ by default if not specified. Our approach requires no modification to existing network architecture and incurs almost no extra computational cost.

---
**Algorithm 1** Self-Adaptive Training
---
**Require:** Data $\{(\boldsymbol{x}_i, \boldsymbol{y}_i)\}_n$, initial targets $\{\boldsymbol{t}_i\}_n = \{\boldsymbol{y}_i\}_n$, batch size $m$, classifier $f$, $E_s = 60$, $\alpha = 0.9$
1: **repeat**
2:     Fetch mini-batch data $\{(\boldsymbol{x}_i, \boldsymbol{t}_i)\}_m$ at current epoch $e$
3:     **for** $i = 1$ **to** $m$ (in parallel) **do**
4:         $\boldsymbol{p}_i = \text{softmax}(f(\boldsymbol{x}_i))$
5:         **if** $e > E_s$ **then** $\boldsymbol{t}_i = \alpha \times \boldsymbol{t}_i + (1 - \alpha) \times \boldsymbol{p}_i$
6:         $w_i = \max_j \boldsymbol{t}_{i,j}$
7:     **end for**
8:     Update $f$ by SGD on $\mathcal{L}(f) = -\frac{1}{\sum_i w_i} \sum_i w_i \sum_j \boldsymbol{t}_{i,j} \log \boldsymbol{p}_{i,j}$
9: **until** end of training
---

## 2.3    Improved generalization of self-adaptive training under random noise

We consider noise scheme (including noise type and noise level) and model capacity as two factors that affect the generalization of deep networks under random noise. We analyze self-adaptive training by varying one of the two factors while fixing the other.

**Varying noise schemes**    We use ResNet-34 [16] and rerun the same experiments in Figure 1a by replacing ERM with our approach. In Figure 1b, we plot the accuracy curves of models trained with our approach on four corrupted training sets and compare with Figure 1a. We highlight the following observations.

- Our approach mitigates the overfitting issue in deep networks. The accuracy curves on noisy training sets (i.e., the red dashed curves in Figure 1b) nearly converge to the percentage of clean data in the training sets, and do not reach perfect accuracy.
- The generalization errors of self-adaptive training (the gap between the red and blue dashed curves in Figure 1b) are much smaller than Figure 1a. We further confirm this observation by displaying the generalization errors of the models trained on the four noisy training sets under various noise rates in the leftmost subfigure of Figure 2. Generalization errors of ERM consistently grow as we increase the injected noise level. In contrast, our approach significantly reduces the generalization errors across all noise levels from 0% (no noise) to 90% (overwhelming noise).
- The accuracy on the clean sets (cyan and yellow solid curves in Figure 1b) is monotonously increasing and converges to higher values than their correspondence in Figure 1a. We also show

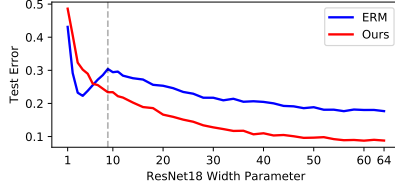
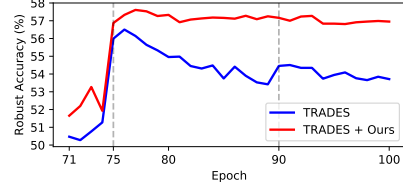

Figure 3: Double-descent ERM *vs.* single-descent self-adaptive training on the error-capacity curve. The model of width 64 corresponds to standard ResNet-18. The vertical dashed line represents the interpolation threshold [5, 24].

Figure 4: Robust Accuracy (%) on CIFAR10 test set under white box $\ell_\infty$ PGD-20 attack ($\epsilon$=0.031). The vertical dashed lines indicate learning rate decay.

the clean validation errors in the right two subfigures in Figure 2. The figures show that the error of self-adaptive training is consistently much smaller than that of ERM.

**Varying model capacity** We notice that such analysis is related to a recent-discovered intriguing phenomenon [5, 24] in modern machine learning models: as the capacity of model increases, the test error initially decreases, then increases, and finally shows a second descent. This phenomenon is termed *double descent* [5] and has been widely observed in deep networks [24]. To evaluate the double-descent phenomenon on self-adaptive training, we follow exactly the same experimental settings as [24]: we vary the width parameter of ResNet-18 [16] and train the networks on the CIFAR10 dataset with 15% training label being corrupted at random (details are given in Appendix A.1). Figure 3 shows the curves of test error. It shows that self-adaptive training overall achieves much lower test error than that of ERM in most cases. Besides, we observe that the curve of ERM clearly exhibits the double-descent phenomenon, while the curve of our approach is monotonously decreasing as the model capacity increases. Since the double-descent phenomenon may vanish when label noise is absent [24], our experiment indicates that this phenomenon may be a result of overfitting of noise and we can bypass it by a proper design of training process such as the self-adaptive training.

**Potential failure scenarios** We notice that self-adaptive training could perform worse than ERM when using extremely small models that underfit the training data. Under such cases, the models do not have enough capacity to capture sufficient information, incorporating their ambiguous prediction may even hinder the training dynamics. However, as shown in Figure 3, the ERM can only outperform our self-adaptive training in some extreme cases that the models are $10\times$ smaller than the standard ResNet-18, indicating that our method can work well in most realistic settings.

## 2.4 Improved generalization of self-adaptive training under adversarial noise

Adversarial noise [39] is different from the random noise in that the noise is model-dependent and imperceptible to humans. We use the state-of-the-art adversarial training algorithm TRADES [48] as our baseline to evaluate the performance of self-adaptive training under adversarial noise. Algorithmally, TRADES minimizes

$$\mathbb{E}_{\boldsymbol{x},\boldsymbol{y}}\left\{\mathrm{CE}(\boldsymbol{p}(\boldsymbol{x}),\boldsymbol{y}) + \max_{\|\widetilde{\boldsymbol{x}}-\boldsymbol{x}\|_\infty \leq \epsilon} \mathrm{KL}(\boldsymbol{p}(\boldsymbol{x}),\boldsymbol{p}(\widetilde{\boldsymbol{x}}))/\lambda\right\}, \tag{2}$$

where $\boldsymbol{p}(\cdot)$ is the model prediction, $\epsilon$ is the maximal allowed perturbation, CE stands for the cross entropy, KL stands for the Kullback–Leibler divergence, and the hyper-parameter $\lambda$ controls the trade-off between robustness and accuracy. We replace the CE term in TRADES loss with our method. The models are evaluated using robust accuracy $\frac{1}{n}\sum_i \mathbb{1}\{\mathrm{argmax}\ \boldsymbol{p}(\widetilde{\boldsymbol{x}}_i) = \mathrm{argmax}\ \boldsymbol{y}_i\}$, where adversarial example $\widetilde{\boldsymbol{x}}$ are generated by white box $\ell_\infty$ projected gradient descent (PGD) attack [22] with $\epsilon = 0.031$, perturbation steps of 20. We set the initial learning rate as 0.1 and decay it by a factor of 0.1 in epochs 75 and 90, respectively. We choose $1/\lambda = 6.0$ as suggested by [48] and use $\mathrm{E}_s = 70$, $\alpha = 0.9$ for our approach. Experimental details are given in Appendix A.2.

We display the robust accuracy on CIFAR10 test set after $\mathrm{E}_s = 70$ epochs in Figure 4. It shows that the robust accuracy of TRADES reaches its highest value around the epoch of first learning rate decay (epoch 75) and decreases later, which suggests that overfitting might happen if we train the model without early stopping. On the other hand, self-adaptive training considerably mitigates the overfitting issue in the adversarial training and consistently improves the robust accuracy of TRADES

Table 1: Test Accuracy (%) on CIFAR datasets with various levels of uniform label noise injected to training set. We compare with previous works under exactly the same experiment settings. It shows that in most settings, self-adaptive training improves over the state-of-the-art as significant as 9%.

| Backbone | Label Noise Rate | CIFAR10 | | | | CIFAR100 | | | |
|---|---|---|---|---|---|---|---|---|---|
| | | 0.2 | 0.4 | 0.6 | 0.8 | 0.2 | 0.4 | 0.6 | 0.8 |
| ResNet-34 | ERM + Early Stopping | 85.57 | 81.82 | 76.43 | 60.99 | 63.70 | 48.60 | 37.86 | 17.28 |
| | Label Smoothing [38] | 85.64 | 71.59 | 50.51 | 28.19 | 67.44 | 53.84 | 33.01 | 9.74 |
| | Forward $\hat{T}$ [29] | 87.99 | 83.25 | 74.96 | 54.64 | 39.19 | 31.05 | 19.12 | 8.99 |
| | Mixup [47] | 93.58 | 89.46 | 78.32 | 66.32 | 69.31 | 58.12 | 41.10 | 18.77 |
| | Trunc $\mathcal{L}_q$ [49] | 89.70 | 87.62 | 82.70 | 67.92 | 67.61 | 62.64 | 54.04 | 29.60 |
| | Joint Opt [40] | 92.25 | 90.79 | 86.87 | 69.16 | 58.15 | 54.81 | 47.94 | 17.18 |
| | SCE [43] | 90.15 | 86.74 | 80.80 | 46.28 | 71.26 | 66.41 | 57.43 | 26.41 |
| | DAC [42] | 92.91 | 90.71 | 86.30 | 74.84 | 73.55 | 66.92 | 57.17 | 32.16 |
| | SELF [25] | - | 91.13 | - | 63.59 | - | 66.71 | - | 35.56 |
| | Ours | **94.14** | **92.64** | **89.23** | **78.58** | **75.77** | **71.38** | **62.69** | **38.72** |
| WRN28-10 | ERM + Early Stopping | 87.86 | 83.40 | 76.92 | 63.54 | 68.46 | 55.43 | 40.78 | 20.25 |
| | MentorNet [17] | 92.0 | 89.0 | - | 49.0 | 73.0 | 68.0 | - | 35.0 |
| | DAC [42] | 93.25 | 90.93 | 87.58 | 70.80 | 75.75 | 68.20 | 59.44 | 34.06 |
| | SELF [25] | - | **93.34** | - | 67.41 | - | 72.48 | - | 42.06 |
| | Ours | **94.84** | 93.23 | **89.42** | **80.13** | **77.71** | **72.60** | **64.87** | **44.17** |

by 1%∼3%, which indicates that our method can improve the generalization in the presence of adversarial noise.

# 3 Application I: Classification with Label Noise

Given improved generalization of self-adaptive training over ERM under noise, we provide applications of our approach which outperforms the state-of-the-art with a significant gap.

## 3.1 Problem formulation

Given a set of noisy training data $\{(\boldsymbol{x}_i, \widetilde{\boldsymbol{y}}_i)\}_n \in \widetilde{\mathcal{D}}$, where $\widetilde{\mathcal{D}}$ is the distribution of noisy data and $\widetilde{\boldsymbol{y}}_i$ is the noisy label for each uncorrupted sample $\boldsymbol{x}_i$, the goal is to be robust to the label noise in the training data and improve the classification performance on clean test data that are sampled from clean distribution $\mathcal{D}$.

## 3.2 Experiments on CIFAR datasets

**Setup**  We consider the case that the labels are assigned uniformly at random with different noise rates. Following prior works [49, 42], we conduct the experiments on the CIFAR10 and CIFAR100 datasets [18] using ResNet-34 [16] and Wide ResNet 28 [45] as our base classifiers. The networks are implemented on PyTorch [28] and optimized using SGD with initial learning rate of 0.1, momentum of 0.9, weight decay of 0.0005, batch size of 256, total training epochs of 200. The learning rate is decayed to zero using cosine annealing schedule [21]. We use data augmentation of random horizontal flipping and cropping. We report the average performance over 3 trials.

**Main results**  We summarize the experiments in Table 1. Most of the results are cited from original papers when they are under the same experiment settings; the results of Label Smoothing [38], Mixup [47], Joint Opt [40] and SCE [43] are reproduced by rerunning the official open-sourced implementations. From the table, we can see that our approach outperforms the state-of-the-art methods in most entries by 1% ∼ 9% on both CIFAR10 and CIFAR100 datasets. Notably, unlike Joint Opt, DAC and SELF methods that require multiple iterations of training, our method enjoys the same computational budget as ERM.

**Ablation study and parameter sensitivity**  First, we report the performance of ERM equipped with simple early stopping scheme in the first row of Table 1. We observe that our approach achieves substantial improvements over this baseline. This demonstrates that simply early stopping the

Table 2: Ablation study on CIFAR datasets in terms of classification Accuracy (%).

(a) Influence of the two components of our approach.

| | CIFAR10 | | CIFAR100 | |
| Noise Rate | 0.4 | 0.8 | 0.4 | 0.8 |
|---|---|---|---|---|
| Ours | **92.64** | **78.58** | **71.38** | **38.72** |
| - Re-weighting | 92.49 | 78.10 | 69.52 | 36.78 |
| - Moving Average | 72.00 | 28.17 | 50.93 | 11.57 |

(b) Parameters sensitivity when label noise of 40% is injected to CIFAR10 training set.

| $\alpha$ | 0.6 | 0.8 | 0.9 | 0.95 | 0.99 |
|---|---|---|---|---|---|
| Fix $E_s$=60 | 90.17 | 91.91 | **92.64** | 92.54 | 84.38 |
| $E_s$ | 20 | 40 | 60 | 80 | 100 |
| Fix $\alpha$=0.9 | 89.58 | 91.89 | **92.64** | 92.26 | 88.83 |

training process is a sub-optimal solution. Then, we further report the influences of two individual components of our approach: Exponential Moving Average (EMA) and sample re-weighting scheme. As displayed in Table 2a, removing any component considerably hurts the performance under all noise rates and removing EMA scheme leads to a significant performance drop. This suggests that properly incorporating model predictions is important in our approach. Finally, we analyze the sensitivity of our approach to the parameters $\alpha$ and $E_s$ in Table 2b (and also Table 5 of Appendix). The performance is stable for various choices of $\alpha$ and $E_s$, indicating that our approach is insensitive to the hyper-parameter tuning.

## 3.3 Experiments on ImageNet dataset

The work of [35] suggested that ImageNet dataset [8] contains annotation errors even after several rounds of cleaning. Therefore, in this subsection, we use ResNet-50 [16] to evaluate self-adaptive training on the ImageNet under both standard setup (i.e., using original labels) and the case that 40% training labels are corrupted. We provide the experimental details in Appendix A.3 and report model performance on the ImageNet validation set in terms of top1 accuracy in Table 3. We see that self-adaptive training consistently improves the ERM baseline by a considerable margin (e.g., 2% when 40% labels are corrupted), which validates the effectiveness of our approach on large-scale dataset.

Table 3: Top1 Accuracy (%) on ImageNet validation set.

| Noise Rate | 0.0 | 0.4 |
|---|---|---|
| ERM | 76.8 | 69.5 |
| Ours | **77.2** | **71.5** |

## 3.4 Further inspection on self-adaptive training

**Label recovery** We demonstrate that our approach is able to recover the true labels from noisy training labels: we obtain the recovered labels by the moving average targets $t_i$ and compute the recovered accuracy as $\frac{1}{n}\sum_i \mathbb{1}\{\text{argmax } y_i = \text{argmax } t_i\}$, where $y_i$ is the clean label of each training sample. When 40% label are corrupted in the CIFAR10 and ImageNet training set, our approach successfully corrects a huge amount of labels and obtains recovered accuracy of 94.6% and 81.1%, respectively. We also display the confusion matrix of recovered labels w.r.t the clean labels on CIFAR10 in Figure 5, from which we see that our approach performs impressively well for all classes.

**Sample weights** Following the same procedure, we display the average sample weights in Figure 6. In the figure, the $(i, j)$-th block contains the average weight of samples with clean label $i$ and recovered label $j$, the white areas represent the case that no sample lies in the cell. We see that the weights on the diagonal blocks are clearly higher than those on non-diagonal blocks. The figure indicates that, aside from impressive ability to recover the correct labels, self-adaptive training could properly down-weight the noisy examples.

## 4 Application II: Selective Classification

### 4.1 Problem formulation

Selective classification, a.k.a. classification with rejection, trades classifier coverage off against accuracy [10], where the coverage is defined as the fraction of classified samples in the dataset; the classifier is allowed to output "don't know" for certain samples. The task focuses on noise-free setting and allows classifier to abstain on potential out-of-distribution samples or samples lies in the tail of

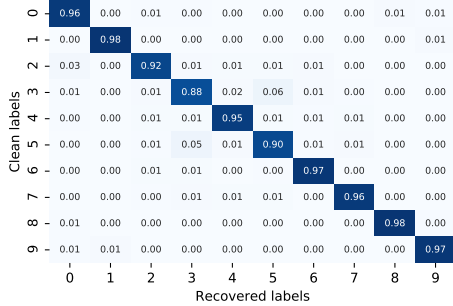
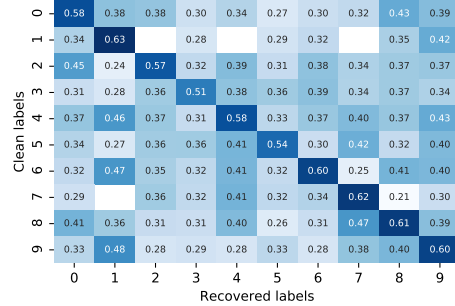

Figure 5: Confusion matrix of recovered labels w.r.t clean labels on CIFAR10 training set with 40% of label noise.

Figure 6: Average sample weights $w_i$ under various labels. The white areas indicate that no sample lies in the cell.

data distribution, that is, making prediction only on samples with confidence. Formally, a selective classifier is a composition of two functions $(f, g)$, where $f$ is the conventional $c$-class classifier and $g$ is the selection function that reveals the underlying uncertainty of inputs. Given an input $\boldsymbol{x}$, selective classifier outputs

$$(f, g)(\boldsymbol{x}) = \begin{cases} \text{Abstain,} & g(\boldsymbol{x}) > \tau; \\ f(\boldsymbol{x}), & \text{otherwise,} \end{cases} \tag{3}$$

for a given threshold $\tau$ that controls the trade-off.

## 4.2 Approach

Inspired by [42, 20], we adapt our presented approach in Algorithm 1 to the selective classification task. We introduce an extra $(c + 1)$-th class (represents *abstention*) during training and replace selection function $g(\cdot)$ in Equation (3) by $f(\cdot)_c$. In this way, we can train a selective classifier in an end-to-end fashion. Besides, unlike previous works that provide no explicit signal for learning abstention class, we use model predictions as a guideline in the design of learning process. Given a mini-batch of data pairs $\{(\boldsymbol{x}_i, \boldsymbol{y}_i)\}_m$, model predictions $\boldsymbol{p}_i$ and its exponential moving average $\boldsymbol{t}_i$ for each sample, we optimize the classifier $f$ by minimizing:

$$\mathcal{L}(f) = -\frac{1}{m} \sum_i [\boldsymbol{t}_{i,y_i} \log \boldsymbol{p}_{i,y_i} + (1 - \boldsymbol{t}_{i,y_i}) \log \boldsymbol{p}_{i,c}], \tag{4}$$

where $y_i$ is the index of non-zero element in the one hot label vector $\boldsymbol{y}_i$. The first term measures the cross-entropy loss between prediction and original label $\boldsymbol{y}_i$, in order to learn a good multi-class classifier. The second term acts as the selection function, identifies uncertain samples in datasets. $\boldsymbol{t}_{i,y_i}$ dynamically trades-off these two terms: if $\boldsymbol{t}_{i,y_i}$ is very small, the sample is deemed as uncertain and the second term enforces the selective classifier to learn to abstain this sample; if $\boldsymbol{t}_{i,y_i}$ is close to 1, the loss recovers the standard cross entropy minimization and enforces the selective classifier to make perfect prediction.

## 4.3 Experiments

We conduct the experiments on two datasets: CIFAR10 [18] and Dogs vs. Cats [1]. We compare our method with previous state-of-the-art methods on selective classification, including Deep Gamblers [20], SelectiveNet [13], Softmax Response (SR) and MC-dropout [12]. We use the same experimental settings as these works for fair comparison (details are given in Appendix A.4). The results of prior methods are cited from original papers and are summarized in Table 4. We see that our method achieves up to 50% relative improvements compared with all other methods under various coverage rates, on all datasets. Notably, Deep Gamblers also introduces an extra abstention class in their method but without applying model predictions. The improved performance of our method comes from the use of model predictions in the training process.

Table 4: Selective classification error rate (%) on CIFAR10 and Dogs vs. Cats datasets for various coverage rates (%). Mean and standard deviation are calculated over 3 trials. The best entries and those overlap with them are marked bold.

| Dataset | Coverage | Ours | Deep Gamblers | SelectiveNet | SR | MC-dropout |
|---------|----------|------|---------------|--------------|-----|------------|
| CIFAR10 | 100 | 6.05±0.20 | 6.12±0.09 | 6.79±0.03 | 6.79±0.03 | 6.79±0.03 |
| | 95 | **3.37±0.05** | **3.49±0.15** | 4.16±0.09 | 4.55±0.07 | 4.58±0.05 |
| | 90 | **1.93±0.09** | 2.19±0.12 | 2.43±0.08 | 2.89±0.03 | 2.92±0.01 |
| | 85 | **1.15±0.18** | **1.09±0.15** | 1.43±0.08 | 1.78±0.09 | 1.82±0.09 |
| | 80 | **0.67±0.10** | **0.66±0.11** | 0.86±0.06 | 1.05±0.07 | 1.08±0.05 |
| | 75 | **0.44±0.03** | 0.52±0.03 | **0.48±0.02** | 0.63±0.04 | 0.66±0.05 |
| | 70 | **0.34±0.06** | 0.43±0.07 | **0.32±0.01** | 0.42±0.06 | 0.43±0.05 |
| Dogs vs. Cats | 100 | 3.01±0.17 | 2.93±0.17 | 3.58±0.04 | 3.58±0.04 | 3.58±0.04 |
| | 95 | **1.25±0.05** | **1.23±0.12** | 1.62±0.05 | 1.91±0.08 | 1.92±0.06 |
| | 90 | **0.59±0.04** | **0.59±0.13** | 0.93±0.01 | 1.10±0.08 | 1.10±0.05 |
| | 85 | **0.25±0.11** | 0.47±0.10 | 0.56±0.02 | 0.82±0.06 | 0.78±0.06 |
| | 80 | **0.15±0.06** | 0.46±0.08 | 0.35±0.09 | 0.68±0.05 | 0.55±0.02 |

## 5   Related Works

**Generalization of deep networks**   Previous work [46] systematically analyzed the capability of deep networks to overfit random noise. Their results show that traditional wisdom fails to explain the generalization of deep networks. Another line of works [26, 27, 2, 37, 5, 14, 24] observed an intriguing double-descent risk curve from the bias-variance trade-off. [5, 24] claimed that this observation challenges the conventional U-shaped risk curve in the textbook. Our work shows that this observation may stem from overfitting of noise; the phenomenon vanishes by a proper design of training process such as self-adaptive training. To improve the generalization of deep networks, [38, 30] proposed label smoothing regularization that uniformly distributes $\epsilon$ of labeling weight to all classes and uses this soft label for training; [47] introduced mixup augmentation that extends the training distribution by dynamic interpolations between random paired input images and the associated targets during training. This line of research is similar with ours as both methods use soft labels in the training. However, self-adaptive training is able to recover true labels from noisy labels and is more robust to noise.

**Robust learning from corrupted data**   Aside from the approaches that have been discussed in the last paragraph of Section 1.1, there have also been many other works on learning from noisy data. To name a few, [3, 19] showed that deep neural networks tend to fit clean samples first and overfitting of noise occurs in the later stage of training. [19] further proved that early stopping can mitigate the issues that are caused by label noise. [31, 9] incorporated model predictions into training by simple interpolation of labels and model predictions. We demonstrate that our exponential moving average and sample re-weighting schemes enjoy superior performance. Other works [49, 43] proposed alternative loss functions to cross entropy that are robust to label noise. They are orthogonal to ours and are ready to cooperate with our approach as shown in Appendix B.4. Beyond the corrupted data setting, recent works [11, 44] propose self-training scheme that also uses model predictions as training target. However, they suffers from the heavy cost of multiple iterations of training, which is avoided by our approach.

## 6   Conclusion

In this paper, we study the generalization of deep networks. We analyze the standard training dynamic using ERM and characterize its intrinsic failure cases under data corruptions. Our observations motivate us to propose Self-Adaptive Training—a new training algorithm that incorporates model predictions into training process. We demonstrate that our approach improves the generalization of deep networks under various kinds of corruptions. Finally, we present two applications of self-adaptive training on classification with label noise and selective classification, where our approach significantly advances the state-of-the-art.

## Broader Impact

Our work advances robust learning from data under potential corruptions, which is a common feature for real-world, uncurated large-scale datasets due to the error-prone nature of data acquisition. In contrast to a large existing literature focuses on noisy label setting, our motivation is to provide a generic algorithm that not only is robust to various kinds of noise with varying noise levels, but also incurs no extra computational cost. In practice, these factors are crucial since the exact noise scheme is unknown and the computation budget is indeed very limited. Built upon the analysis on the intrinsic failure patterns of ERM under data corruptions, we introduce an elegant way to incorporate model predictions into training process to improve the generalization of deep networks under noisy data. By correcting outliers and calibrating the training process, our approach is ready for real-world application of deep learning. It can serve as a basic building block of large-scale AI system that generalizes well to a wide range of visual tasks.

While we have empirically evaluated our approach on data under both random and adversarial noise, most of our studies focus on artificial data corruptions (except the analysis on ImageNet which contains annotation error by itself), which may not represent natural noise in practice. However, the presented methodology that properly incorporates model predictions into training process sheds light on understanding and improving the generalization of deep networks under data corruptions in the future study.

## Acknowledgments and Disclosure of Funding

Lang Huang and Chao Zhang were supported in part by the National Nature Science Foundation of China under Grant 62071013 and 61671027, and the National Key R&D Program of China under Grant 2018AAA0100300. Hongyang Zhang was supported in part by the Defense Advanced Research Projects Agency under cooperative agreement HR00112020003. The views expressed in this work do not necessarily reflect the position or the policy of the Government and no official endorsement should be inferred. Approved for public release; distribution is unlimited.

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
