[Supplementary Material]



(a) Accuracy curves of model trained using ERM.

(b) Accuracy curves of model trained using our method.

Figure 7: Accuracy curves of model trained on noisy CIFAR10 training set with 80% noise rate. The horizontal dotted line displays the percentage of clean data in the training sets. It shows that our observations in Section 2 hold true even when extreme label noise injected.

## A  Experimental Setups

### A.1  Double descent phenomenon

Following previous work [12], we optimize all models using Adam [7] optimizer with fixed learning rate of 0.0001, batch size of 128, common data augmentation, weight decay of 0 for 4,000 epochs. For our approach, we use the hyper-parameters $E_s = 40, \alpha = 0.9$ for standard ResNet-18 (width of 64) and dynamically adjust them for other models according to the relation of model capacity $r = \frac{64}{\text{width}}$ as:

$$E_s = 40 \times r; \quad \alpha = 0.9^{\frac{1}{r}}. \tag{1}$$

### A.2  Adversarial training

[17] reported that imperceptible small perturbations around input data (i.e., adversarial examples) can cause ERM trained deep neural networks to make arbitrary predictions. Since then, a large literature devoted to improving the adversarial robustness of deep neural networks. Among them, adversarial training algorithm TRADES [21] achieves state-of-the-art performance. TRADES decomposed robust error (w.r.t adversarial examples) to sum of natural error and boundary error, and proposed to minimize:

$$\mathbb{E}_{\boldsymbol{x},\boldsymbol{y}} \left\{ \text{CE}(\boldsymbol{p}(\boldsymbol{x}), \boldsymbol{y}) + \max_{\|\widetilde{\boldsymbol{x}} - \boldsymbol{x}\|_\infty \leq \epsilon} \text{KL}(\boldsymbol{p}(\boldsymbol{x}), \boldsymbol{p}(\widetilde{\boldsymbol{x}}))/\lambda \right\}, \tag{2}$$

where $\boldsymbol{p}(\cdot)$ is the model prediction, $\epsilon$ is the maximal allowed perturbation, CE stands for cross entropy, KL stands for Kullback–Leibler divergence. The first term corresponds to ERM that maximizes the natural accuracy; the second term pushes the decision boundary away from data points to improve adversarial robustness; the hyper-parameter $1/\lambda$ controls the trade-off between natural accuracy and adversarial robustness. We evaluate self-adaptive training on this task by replacing the first term of Equation (2) with our approach.

Our experiments are based on the official open-sourced implementation[1] of TRADES [21]. Concretely, we conduct experiments on CIFAR10 dataset [8] and use WRN-34-10 [19] as base classifier. For training, we use initial learning rate of 0.1, batch size of 128, 100 training epochs. The learning rate is decayed at 75-th, 90-th epoch by a factor of 0.1. The adversarial example $\widetilde{\boldsymbol{x}}_i$ is generated dynamically during training by projected gradient descent (PGD) attack [11] with maximal $\ell_\infty$ perturbation $\epsilon$ of 0.031, perturbation step size of 0.007, number of perturbation steps of 10. The hyper-parameter $1/\lambda$ of TRADES is set to 6 as suggested by original paper, $E_s, \alpha$ of our approach is set to 70,

Figure 8: Accuracy curves on different portions of the CIFAR10 training set (with 40% label noise) w.r.t. correct labels. We split the training set into two portions: 1) *Untouched portion*, i.e., the elements in the training set which were left untouched; 2) *Corrupted portion*, i.e., the elements in the training set which were indeed randomized. It shows that ERM fits correct labels in the first few epochs and then eventually overfits the corrupted labels. In contrast, self-adaptive training calibrates the training process and consistently fits the correct labels.

0.9, respectively. For evaluation, we report robust accuracy $\frac{1}{n}\sum_i \mathbb{1}\{\mathrm{argmax}\ p(\widetilde{\boldsymbol{x}}_i) = \mathrm{argmax}\ \boldsymbol{y}_i\}$, where adversarial example $\widetilde{\boldsymbol{x}}$ is generated by white box $\ell_\infty$ untargeted PGD attack with $\epsilon$ of 0.031, perturbation step size of 0.007, number of perturbation steps of 20.

### A.3 ImageNet

We use ResNet-50 [4] as base classifier. Following original paper [4] and [10, 2], we use SGD to optimize the networks with batch size of 768, base learning rate of 0.3, momentum of 0.9, weight decay of 0.0005 and total training epoch of 95. The learning rate is linearly increased from 0.0003 to 0.3 in first 5 epochs (i.e., warmup), and then decayed using cosine annealing schedule [10] to 0. Following common practice, we use random resizing, cropping and flipping augmentation during training. The hyper-parameters of our approach are set to $\mathrm{E}_s = 50$ and $\alpha = 0.99$ under standard setup, and are set to $\mathrm{E}_s = 60$ and $\alpha = 0.95$ under 40% label noise setting. The experiments are conducted on PyTorch [13] with distributed training and mixed precision training[2] for acceleration.

### A.4 Selective classification

The experiments are base on official open-sourced implementation[3] of Deep Gamblers to ensure fair comparison. We use the VGG-16 network [15] with batch normalization [6] and dropout [16] as base classifier in all experiments. The network is optimized using SGD with initial learning rate of 0.1, momentum of 0.9, weight decay of 0.0005, batch size of 128, total training epoch of 300. The learning rate is decayed by 0.5 in every 25 epochs. For our method, we set the hyper-parameters $\mathrm{E}_s = 0, \alpha = 0.99$.

## B  Additional Experimental Results & Discussions

### B.1  ERM may suffer from overfitting of noise

In [20], the authors showed that the model trained by standard ERM can easily fit randomized data. However, they only analyzed the generalization errors in the presence of corrupted labels. In this paper, we report the whole training process and also consider the performance on clean sets (i.e., the original uncorrupted data). Figure 1a shows the four accuracy curves (on clean and noisy training, validation set, respectively) for each model that is trained on one of four corrupted training data. Note that the models can only have access to the noisy training sets (i.e., the red curve) and the other three curves are shown only for the illustration purpose. We conclude with two principal observations from the figures: (1) The accuracy on noisy training and validation sets is close at beginning and the gap is monotonously increasing w.r.t. epoch. The generalization errors (i.e., the gap between the accuracy on noisy training and validation sets) are large at the end of training. (2) The accuracy on

Figure 9: Generalization error and clean validation error under four kinds of random noise (represented by different colors) for ERM (the dashed curves) and our approach (the solid curves) on CIFAR10 when data augmentation is turned off. We zoom-in the dashed rectangle region and display it in the third column for clear demonstration.

Figure 10: Self-adaptive training *vs.* ERM on the error-epoch curve. We train the standard ResNet-18 networks (i.e., width of 64) on the CIFAR10 dataset with 15% randomly-corrupted labels and report the test errors on the clean data. The dashed vertical line represents the initial epoch $E_s$ of our approach. It shows that self-adaptive training has significantly diminished epoch-wise double-descent phenomenon.

clean training and validation set is consistently higher than the percentage of clean data in the noisy training set. This occurs around the epochs between underfitting and overfitting.

Our first observation poses concerns on the overfitting issue of ERM training dynamic which has also been reported by [9]. However, the work of [9] only considered the case of corrupted labels and proposed using early-stop mechanism to improve the performance on clean data. On the other hand, our analysis of the broader corruption schemes shows that the early stopping might be sub-optimal and may hurt the performance under other types of corruptions (see the last three columns in Figure 1a).

The second observation implies that, perhaps surprisingly, model predictions by ERM can capture and amplify useful signals in the noisy training set, although the training dataset is heavily corrupted. While this was also reported in [20, 14, 3, 9] for the case of corrupted labels, we show that similar phenomenon occurs under other kinds of corruptions more generally. This observation sheds light on our approach, which incorporates model predictions into training procedure.

## B.2 Improved generalization of self-adaptive training on random noise

**Training accuracy w.r.t. correct labels on different portions of data**   For more intuitive demonstration, we split the CIFAR10 training set (with 40% label noise) into two portions: 1) *Untouched portion*, i.e., the elements in the training set which were left untouched; 2) *Corrupted portion*, i.e., the elements in the training set which were indeed randomized. The accuracy curves on these two portions w.r.t correct training labels is shown in Figure 8. We can observe that the accuracy of ERM on the corrupted portion first increases in the first few epochs and then eventually decreases to 0. In contrast, self-adaptive training calibrates the training process and consistently fits the correct labels.

**Study on extreme noise**   We further rerun the same experiments as in Figure 1 of main text by injecting extreme noise (i.e., noise rate of 80%) into CIFAR10 dataset. We report the corresponding accuracy curves in Figure 7, which shows that our approach significantly improves the generalization over ERM even when random noise dominates training data. This again justify our observations in Section 2 of the main body.

**Effect of data augmentation**   All our previous studies are performed with common data augmentation (i.e., random cropping and flipping). Here, we further report the effect of data augmentation. We adjust introduced hyper-parameters as $E_s = 25$, $\alpha = 0.7$ due to severer overfitting when data

Table 5: Parameters sensitivity to different datasets and noise rates.

| | CIFAR10 (80% Noise) | | | CIFAR100 (40% Noise) | | |
|---|---|---|---|---|---|---|
| $\alpha$ | 0.8 | 0.9 | 0.95 | 0.8 | 0.9 | 0.95 |
| Fix $E_s$=60 | 75.60 | **78.58** | 75.44 | 70.36 | **71.38** | 68.57 |
| $E_s$ | 40 | 60 | 80 | 40 | 60 | 80 |
| Fix $\alpha$=0.9 | 68.27 | 78.58 | **78.65** | 70.30 | **71.38** | 67.32 |

Table 6: Test Accuracy (%) on CIFAR datasets with various levels of uniform label noise injected to training set. We show that considerable gains can be obtained when combined with SCE loss.

| | CIFAR10 | | | | CIFAR100 | | | |
|---|---|---|---|---|---|---|---|---|
| Method | Label Noise Rate | | | | Label Noise Rate | | | |
| | 0.2 | 0.4 | 0.6 | 0.8 | 0.2 | 0.4 | 0.6 | 0.8 |
| SCE [18] | 90.15 | 86.74 | 80.80 | 46.28 | 71.26 | 66.41 | 57.43 | 26.41 |
| Ours | 94.14 | 92.64 | 89.23 | 78.58 | 75.77 | 71.38 | 62.69 | 38.72 |
| Ours + SCE | **94.39** | **93.29** | **89.83** | **79.13** | **76.57** | **72.16** | **64.12** | **39.61** |

augmentation is absent. The Figure 9 shows the corresponding generalization errors and clean validation errors. We observe that, for both ERM and our approach, the errors clearly increase when data augmentation is absent (compared with those in Figure 2). However, the gain is limited and the generalization errors can still be very large, with or without data augmentation for standard ERM. Directly replacing the standard training procedure with our approach can bring bigger gains in terms of generalization regardless of data augmentation. This suggests that data augmentation can help but is not of essence to improve generalization of deep neural networks, which is consistent with the observation in [20].

## B.3 Epoch-wise double descent phenomenon

Prior work [12] reported that, for sufficient large model, test error-training epoch curve also exhibits double-descent phenomenon, which they termed *epoch-wise double descent*. In Figure 10, we reproduce the epoch-wise double descent phenomenon on ERM and inspect self-adaptive training. We observe that our approach (the red curve) exhibits slight double-descent due to overfitting starts before initial $E_s$ epochs. As the training targets being updated (i.e., after $E_s$ = 40 training epochs), the red curve undergoes monotonous decrease. This observation again indicates that double-descent phenomenon may stem from overfitting of noise and can be avoided by our algorithm.

## B.4 Cooperation with symmetric cross entropy

Prior work [18] showed that Symmetric Cross Entropy (SCE) loss is robust to underlying label noise in training data. Formally, given training target $t_i$ and model prediction $p_i$, SCE loss is defined as:

$$\mathcal{L}_{sce} = -w_1 \sum_j t_{i,j} \, \log \, p_{i,j} - w_2 \sum_j p_{i,j} \, \log \, t_{i,j}, \tag{3}$$

where the first term is the standard cross entropy loss and the second term is the reversed version. In this section, we show that self-adaptive training can cooperate with this noise-robust loss and enjoy further performance boost without extra cost.

**Setup** The most experiments settings are kept the same as Section 3.2. For the introduced hyper-parameters $w_1, w_2$ of SCE loss, we directly set them to $1, 0.1$, respectively, in all our experiments for simplicity.

**Results** We summarize the results in Table 6. We cam see that, although self-adaptive training already achieves very strong performance, considerable gains can be obtained when equipped with SCE loss. Concretely, the improvement is as large as 1.5% when label noise of 60% injected to

Table 7: Average Accuracy (%) on CIFAR10 test set and out-of-distribution dataset CIFAR10-C at various corruption levels.

| Method | CIFAR10 | Corruption Level@CIFAR10-C | | | | |
|---|---|---|---|---|---|---|
| | | 1 | 2 | 3 | 4 | 5 |
| ERM | 95.32 | 88.44 | 83.22 | 77.26 | 70.40 | 58.91 |
| Ours | **95.80** | **89.41** | **84.53** | **78.83** | **71.90** | **60.77** |

CIFAR100 training set. It also indicates that our approach is flexible and is ready to cooperate with alternative loss functions.

## B.5 Out-of-distribution generalization

In this section, we consider out-of-distribution (OOD) generalization, where the models are evaluated on unseen test distributions outside the training distribution.

**Setup** To evaluate the OOD generalization performance, we use CIFAR10-C benchmark [5] that constructed by applying 15 types of corruption to the original CIFAR10 test set at 5 levels of severity. The performance is measure by average accuracy over 15 types of corruption. We mainly follow the training details in Section 3.2 and adjust $\alpha = 0.95, \mathrm{E}_s = 80$.

**Results** We summarize the results in Table 7. Regardless the presence of corruption and corruption levels, our method consistently outperforms ERM by a considerable margin, which becomes large when the corruption is more severe. The experiment indicates that self-adaptive training may provides implicit regularization for OOD generalization.

## B.6 Cost of maintaining probability vectors

Take the large-scale ImageNet dataset [1] as an example. The ImageNet consists of about 1.2 million images categorized to 1000 classes. The storage of such vectors in single precision format for the entire dataset requires $1.2 \times 10^6 \times 1000 \times 32$ bit $\approx 4.47$GB, which is acceptable since modern GPUs usually have no less than 11GB memory. Moreover, the vectors can be stored on CPU memory or even disk and loaded along with the images to further reduce the cost.

## Footnotes

[1]`https://github.com/yaodongyu/TRADES`

[2]https://github.com/NVIDIA/apex

[3]https://github.com/Z-T-WANG/NIPS2019DeepGamblers