[Reviews · NeurIPS 2020]

Review 1

Summary and Contributions: This paper proposed a new training algorithm that incorporates model predictions into the training process to solve the overfitting issue and improves generalization under both random and adversarial noises.They first analyze the standard ERM training process and find the useful information for classification has been distilled to model predictions in the first few epochs. And then they show the double descent phenomenon in ERM might be a result of overfitting of noise and their proposed method only has a single-descent curve. Finally, they validate their method in CIFAR and ImageNet with two applications, classification with label noise and selective classification.

Strengths: 1. The idea of combining the model predictions and the noisy label to solve the overfitting issue and improve the generalization is novel and may inspire other researchers in the efficient training or adversarial training field. 2. A detailed ablation study is conducted to demonstrate the effectiveness of each proposed techniques, which make it a convincing framework. 3. The logic flow to demonstrate the proposed self-adaptive training via detailed experiments is clear.

Weaknesses: 1. Lacking insight on setting the hyper-params: Table 2 (b) shows E_s = 60 will have the best performance but in the experiments in Section 2.4, E_s is set to 70. Is there any insight or explanation on the reason why to do such changes? 2. Not enough model structures in the experiments to support the conclusion: As mentioned by one of the baselines [24], previous works utilize a various set of different architectures, which hinders a fair comparison. And this paper only uses ResNet34@CIFAR10/100, and ResNet50@ImageNet. Although this paper reproduces some baselines in the setting above, it would be better to provide more comparison on other model structures and use the reported numbers in the baselines.

Correctness: Yes

Clarity: Yes

Relation to Prior Work: Yes

Reproducibility: Yes

Additional Feedback: More details in the Weaknesses. =====After Rebuttal===== I found my two main concerns has been partly addressed but I still think this work is not stable enough to all the settings in the paper, e.g., 1. in Table 1(a), worse than SELF[24] in CIFAR10@0.4 noise rate but much higher in CIFAR10@0.8 noise rate, and discussion on such phenomena is missing; 2. in Table 1(b) E_s = 60 or 80 has comparable performance when alpha = 0.9 in CIFAR10@80% noise, whlie has a very large gap (~4%) in CIFAR100@40% noise, which will cause more problems when applying this work to other settings not showed in the paper. And the method to decide the hyperparams mentioned in the rebuttal needs to training using TRADES on the experiment setting first, which is not consistent with the efficient training aim in this paper.


Review 2

Summary and Contributions: This work introduces a training method to deal with corrupted data. The algorithm consists in two components. First, the model "bootstraps" its own predictions to correct the potentially wrong labels. Second, the model re-weighs the importance of each sample according to a measure of confidence of the corrected label. Experiments are conducted on CIFAR-01/100 and ImageNet with label noise, and on CIFAR-10 and Dogs vs. Cats for selective prediction.

Strengths: The proposed algorithm is rather simple, both to describe and to implement. It is general and does not incur any additional computational cost. The empirical evaluation indicate that the algorithm provides good results in practice in the presence of noise, on a few relevant image classification tasks.

Weaknesses: The main weakness of the proposed approached is that it is not supported by any theoretical insight. In particular, the success of the method hinges on the premise that the model is able to guess the right predictions so as to correct the noisy labels. Since there is no theoretical criterion to verify that premise, it is not possible to predict whether this proposed method will work well on new learning tasks. Going further, one can imagine cases where this method would fail and actually perform worse than ERM. For instance, if the model is unable to capture sufficient information from the data distribution (for instance if the data distribution is very complex and / or if there are too few training samples and / or if the model does not have sufficient capacity), it would be impossible for the model to "bootstrap" its own predictions and guess the correct labels. It is then likely that self-adaptive training would enter a harmful cycle of making wrong predictions and fitting more and more on these misinformed guesses. In addition, while the method claims to be applicable to any deep supervised learning task, the experiments are only performed on image classification tasks with convolutional neural networks. In the absence of formal guarantees, this makes it difficult to predict whether this claim holds for other modalities. Overall, while the empirical results indicate good performance on the evaluated tasks, I remain concerned about the reliability and robustness of this method as it stands.

Correctness: The empirical results seem to be fairly conducted.

Clarity: The paper is well organised, clear and overall easy to follow.

Relation to Prior Work: To the best of my knowledge, this work provides sufficient comparison with existing work.

Reproducibility: Yes

Additional Feedback: *** Post-rebuttal Comments *** I have carefully read the authors' rebuttal. I agree that it would be unreasonable to require the method to outperform ERM on *any* task. With this said, in the absence of justification or intuition about the reliability of the method, one would still need to demonstrate that it does work on a wide range of models / data modalities / loss functions (potentially also identifying the precise regime where it works well and where it does not). At this time, I remain unconvinced that this has been sufficiently done to claim reliable performance, and thus I am keeping my score (slightly below the acceptance threshold).


Review 3

Summary and Contributions: The paper considers robust learning from corrupted data, for example data subject to label noise or adversarial perturbations. It proposes a self-training approach which progressively averages the model’s predictions with the true (noisy) labels, along with a sample weighting scheme. The self-training approach can be used for a variety of vision tasks, including classification under label noise, adversarial training, and selective classification and achieves state-of-the-art performance on a variety of benchmarks.

Strengths: - The method is conceptually simple and achieves good performance on a range of different tasks and benchmark data sets. In particular, the method seems to mitigate the double-decent phenomenon [5, 23]. - The paper presents a variety of convincing ablation experiments. - In some cases, the proposed method can be combined with other methods to increase robustness, obtaining additional benefits (i.e. the method is in some sense orthogonal to prior approaches).

Weaknesses: - If I understand correctly, the method requires maintaining a probability vector for each data point. This is not an issue for small data sets with few classes, but can become a problem at ImageNet scale. I did not find any comment regarding this issue in the main paper or in the supplement. Could the authors please elaborate on this? - From Table 2 b) it seems that for 40% label noise on CIFAR10 the method is reasonably robust to the hyper parameter values. Does this observation transfer to other corruption percentages and data sets? - Additional experiments on larger data sets would be nice (but I understand that compute might be an issue). --- Thanks for the author response. I still think maintaining the probabilities might become an issue, in particular at large batch size, but I don't think this aspect is critical. Generally, the response addressed my concerns well.

Correctness: As discussed above, the experiments seem carefully executed. Some additional experiments might yield further insights.

Clarity: The paper is largely well-written and easy to follow, and presents sufficient detail on the setup and the experiments.

Relation to Prior Work: The related work section looks fine to me. However, I’m not an expert on the double-descent phenomenon. Some additional references on self-training: Xie et al. "Self-training with noisy student improves imagenet classification." CVPR 2020. Furlanello et al. "Born again neural networks." ICML 2018.

Reproducibility: Yes

Additional Feedback: It would be interesting to investigate the effect of the regularizer on out-of-distribution generalization data sets such as CIFAR10-C/ImageNet-C. Typos: L108: dynamic -> dynamics L156: improves -> improve


Review 4

Summary and Contributions: This paper proposes an improved training objective aimed at reducing overfitting and recovering from label and input noise. The proposed approach modifies the standard objective for training classifiers with two adaptive terms: one term re-weights the ground truth labels according to the current classifier predictions and the other term re-weights training samples according to the confidence of classifier. The approach is simple to implement and incurs no additional cost during training. The authors show that the proposed approach can increase the generalization of the classifier and recover from label noise.

Strengths: This is a well presented and simple approach that shows good empirical performance. Particularly impressive to me are the training label corruption results, in Figure 5 the authors show that the model is able to recover the true labels almost exactly under a 40% noise level. The experiments are complete and show the merit of the proposed approach. Ablation is also conducted over the two hyperparameters. It is straightforward to implement, increasing its potential to be widely adopted.

Weaknesses: For experimental completeness, the uncorrupted labels results on CIFAR10 and CIFAR100 should be reported in Table 1. There is an apparent contradiction in the paper between L30 in which the authors state that the first few training iterations fit the model to correctly predict "easy" labels and L103 where the opposite claim is made (that the first few training iterations are unstable). This should be addressed.

Correctness: The claims in the paper are well supported by the experiments.

Clarity: This paper is well structured and easy to follow.

Relation to Prior Work: Prior work is sufficiently discussed and the proposed work is properly framed

Reproducibility: Yes

Additional Feedback: I would have liked to see more discussion of the sample re-weighting aspect of the work. In particular, I am concerned that with \alpha=0.9, the minimum value for w_i is 0.9. In the sample corruption experiments, was the proposed method able to re-weight the corrupted samples correctly? A similar result in this case as Figure 5 would strengthen the paper. ------- The weaknesses I observed were satisfactorily addressed in the author response. I maintain that this is a good submission.

[Author Response · NeurIPS 2020]

Table 1: Additional experiments in terms of classification Accuracy (%).

(a) Results of WRN28-10@CIFAR and ResNet101@ImageNet.

| Noise Rate | CIFAR10 | | CIFAR100 | | ImageNet |
|---|---|---|---|---|---|
| | 0.4 | 0.8 | 0.4 | 0.8 | 0.0 |
| ERM | 75.41 | 30.00 | 55.68 | 13.99 | 78.2 |
| SELF[24] | **93.34** | 67.41 | 72.48 | 42.06 | - |
| Ours | 93.23 | **80.13** | **72.60** | **44.17** | **78.7** |

(b) Parameters sensitivity to different datasets and noise rates.

| | CIFAR10 (80% Noise) | | | CIFAR100 (40% Noise) | | |
|---|---|---|---|---|---|---|
| $\alpha$ | 0.8 | 0.9 | 0.95 | 0.8 | 0.9 | 0.95 |
| Fix $E_s$=60 | 75.60 | **78.58** | 75.44 | 70.36 | **71.38** | 68.57 |
| $E_s$ | 40 | 60 | 80 | 40 | 60 | 80 |
| Fix $\alpha$=0.9 | 68.27 | 78.58 | **78.65** | 70.30 | **71.38** | 67.32 |

We thank the reviewers for their valuable comments. We summarize major concerns from reviewers and respond to
them appropriately as follows. We will add suggested experiments, references, and fix typos in the updated version.

**To Reviewer #1:** *Q1:* Insights on setting the hyper-parameter $E_s$. *A1:* The
optimal value of $E_s$ is related to the begining epoch of overfitting. Since the
baseline TRADES uses a step learning-rate-decay schedule and we observe that it
starts to overfit around the epoch of the first learning rate decay (the 75-th epoch,
see the blue curve in Figure 1), we simply set the $E_s$ to a slightly smaller value
70. However, we also find that our method is not sensitive to $E_s$: using $E_s = 60$
has similar performance as $E_s = 70$ (see Figure 1, the red and green curves).
This phenomenon is consistent with Table 2b in the main body, where we show
our method is not sensitive to various hyper-parameters (see also Table 1b).

Figure 1: Sensitivity of $E_s$ in the adversarial learning.

*Q2:* Extra experiments on other model structures. *A2:* Besides ResNet34@CIFAR10/100 and ResNet50@ImageNet
in main body, we also report the results of WRN28-10@CIFAR10/100 and ResNet101@ImageNet in Table 1a, where
our approach outperforms ERM and the reported numbers of SELF[24] in most entries, sometimes by a large margin.

**To Reviewer #2:** *Q1:* Claim of applicability to any deep supervised learning task; validity of premise in new tasks.
*A1:* We point out a potential misunderstanding about our paper, where we did NOT claim that our method is applicable
to *any* deep supervised learning task. Though not for any task, our method is robustly and reliably effective for *a wide*
*range of tasks*: classification with label noise, selective classification, adversarial learning, vanishing double descent,
etc. As a new task, we run experiments on OOD generalization task (see Table 2 and A4@R#3), where ours has better
performance. The premise that the model can guess the right predictions follows the observation that deep models fit the
clean samples first, which is justified by [14,18,32] and our extensive experiments in the above-mentioned broad tasks.
*Q2:* Extreme failure cases. *A2:* In Figure 3 of main body, ERM performs better only in the extreme case where
the model capacity is more than $10\times$ smaller than standard ResNet-18, where ERM's test accuracy is poor ($\leq 78\%$,
i.e., significant underfitting occurs). For other extreme cases when the data is complex or when there are few training
samples, though our method might fail, ERM will perform poorly too, due to the information-theoretical limit. We
argue that studying models with enough capacity, realistic (amount of) input data, and reasonable performance (e.g.,
$\geq 90\%$ test accuracy) might be of more interests to the community, where our method consistently outperforms ERM.

**To Reviewer #3:** *Q1:* Cost of maintaining probability vectors. *A1:* The cost is
not high. Take the large-scale ImageNet as an example. The storage of such vectors
in single precision format for the entire dataset requires $1.2 \times 10^6 \times 1000 \times 32$ bit
$\approx 4.47$GB, which is acceptable since modern GPUs usually have no less than 11GB
memory. Moreover, the vectors can be stored on CPU memory or even disk and
loaded along with the images to further reduce the cost.

Table 2: Average Accuracy (%) on CIFAR10-C at various corruption levels.

| Level | 1 | 3 | 5 |
|---|---|---|---|
| ERM | 88.44 | 77.26 | 58.91 |
| Ours | **89.41** | **78.83** | **60.77** |

*Q2:* Robustness to other corruption percentages and datasets. *A2:* In Table 1b, we
conduct extra experiments on two datasets with different noise rates. The results indicates that our approach is robust to
the values of hyper-parameters in various settings.
*Q3:* Additional large-scale experiments. *A3:* Please refer to Table 1a and A2@R#1 for additional results on ImageNet.
*Q4:* OOD generalization. *A4:* In Table 2, we report the average accuracy using ResNet-34 on CIFAR10-C over 15
corruptions. Under various corruption levels, our method consistently outperforms ERM by a considerable margin,
indicating that self-adaptive training provides implicit regularization for OOD generalization.

**To Reviewer #4:** *Q1:* Results on uncorrupted CIFAR. *A1:* On CIFAR10/100,
the test accuracy is 95.32%/78.42% for ERM, and 95.17%/78.69% for ours.
*Q2:* Contradiction between L30 and L103. *A2:* We will make it clear: in
the first few iterations, though the model learns to fit the correct labels in a
*progressive* manner (as in L30), its predictions are very unstable, especially in
the very beginning of the training procedure (as in L103). The instability is due
to the use of regularization such as data augmentation (as in L104).
*Q3:* Further investigation of sample weights. *A3:* The minimum value of $w_i$
is not bounded by $\alpha$ due to the moving-average scheme that accumulates the
predictions. Following the procedure in Figure 5 of main body, we display the
average sample weights in Figure 2, where the white areas represent the case that
no sample lies in the cell. We see that the weights on the diagonal are higher.

Figure 2: Average sample weights $w_i$ under various labels.

[Meta-Review · NeurIPS 2020]

The paper focuses on the problem of learning from corrupted data (e.g. label noise) and introduces an improved training objective. This objective can be interpreted as a self-training whereby the model's predictions are progressively averaged with the true (and possibly noisy labels) coupled with a sample weighting scheme which improves training stability. The authors show that this approach can be used for a variety of vision tasks, including classification under label noise, adversarial training, and selective classification. The reviewers appreciated the conceptual simplicity of the method, the clarity of the presentation, and the promising empirical results. The discussion phase focused on the following two drawbacks: - Theoretical justification: While the theoretical analysis is hard for the general case, it might be doable in the corrupted linear regression case, which could offer some valuable insights. There could be cases where such a scheme performs worse that ERM, and this should be discussed in the manuscript. The reviewers' opinions remain split on this issue. - Empirical validation: Given that the experiments were performed only in the vision domain and a small number of models it is hard to judge whether this approach will robustly generalize to other modalities. Notwithstanding the criticism above, the paper provides a relatively novel and conceptually simple idea which is supported by solid experiments on a topic relevant to the broader NeurIPS community, and I will recommend acceptance. I strongly advise the authors to include all the relevant information from the rebuttal, and prominently display the potential failure models of the proposed method.